


**Deep learning of flood forecasting by considering interpretability**
**and physical constraints**
Ting Zhang *, Ran Zhang, Jianzhu Li, Ping Feng
State Key Laboratory of Hydraulic Engineering Intelligent Construction and Operation, Tianjin
University, Tianjin 300072, China
Corresponding author: Ting Zhang (zhangting_hydro@tju.edu.cn)
**ABSTRACT**
Deep learning models have been proven to be effective in flood forecasting by leveraging the
rich time-series information in the data. However, their limited interpretability and lack of physical
mechanisms remain significant challenges. To address these limitations, this study introduces a
novel model called PHY-FTMA-LSTM, which combines the feature-time-based multi-head
attention mechanism with physical constraints. The PHY-FTMA-LSTM model takes four essential
features of runoff, rainfall, evapotranspiration, and initial soil moisture as inputs to forecast floods
in the Luan River Basin with a lead time of 1-6 h. It emphasizes the significance of relevant factors
in the input features and historical moments through the feature-time attention module. Furthermore,
the model enhances physical consistency by considering the monotonic relationship between the
input variables and the output results. The results demonstrate that the PHY-FTMA-LSTM in most
cases outperforms the original LSTM, the feature-time-based attention LSTM (FTA-LSTM), and
the feature-time-based multi-head attention LSTM (FTMA-LSTM). For a lead time of t+1, the
model achieves an NSE of 0.988, with KGE and $R^2$ of 0.984 and 0.988. The NSE, KGE, and $R^2$ also
reach 0.908, 0.905, and 0.911 for a lead time of t+6. The proposed PHY-FTMA-LSTM model
achieves excellent prediction accuracy, offering valuable insights for enhancing interpretability and
physical consistency in deep learning approaches.
**Keywords**:Deep learning;Flood forecasting;Physical constraints;Attention mechanism
**1. Introduction**
Floods are one of the most common and destructive natural hazards, posing a great threat to
human life, infrastructure, and socio-economic conditions (Kellens et al., 2013; Mourato et al.,
2021). Building reliable and accurate flood forecasting models is the foundation for sustainable



flood risk management with a focus on prevention and protection, and is one of the most challenging
tasks in hydrological forecasting (Birkholz et al., 2014; Zhang et al., 2016).
Traditional hydrological models simulate hydrological processes such as rainfall runoff with a
clear physical meaning, but their construction often demands rich hydro-meteorological data and
subsurface information. Additionally, the large number of parameters involved poses challenges in
determining their values, limiting their practical applicability (Chen et al., 2011). In contrast, data-
driven machine learning (ML) models, which do not rely on explicit consideration of the physical
mechanisms governing hydrological processes and only analyze the statistical relationships between
inputs and outputs, have been widely used in hydrology in recent years (Lima et al., 2016; Yang et
al., 2020; Yu et al., 2006; Zhu et al., 2005). Among them, deep learning (DL) models with multiple
hidden layers have demonstrated significant advantages, including convolutional neural networks
(CNNs), recurrent neural networks (RNNs), and their variants such as long short-term memory
neural networks (LSTMs), and gated recurrent units (GRUs). LSTM, a type of RNN, is specifically
designed for learning long-term dependencies, and its architectural enhancements effectively
address issues such as gradient disappearance and explosion that are inherent to traditional RNNs.
Consequently, LSTM has emerged as a highly favored model in flood forecasting (Cui et al., 2021a;
Kao et al., 2020; Luppichini et al., 2022; Lv et al., 2020).
The DL models, with their powerful characterization capabilities, excel in fitting observations
and have high prediction accuracy for hydrological problems such as flood forecasting, but they still
have limitations. First, the interpretability of DL models is poor (Nearing et al., 2021). The inherent
black-box nature of DL models makes it difficult to understand the significance of model parameters
and the decision-making process. The attention mechanism is an approach to enhance the
interpretability of DL models (Vaswani et al., 2017). Attention allows for the interpretation of
feature importance by selectively emphasizing critical information from a multitude of input
variables through attention weights. Moreover, attention weights can be visualized to gain insights
into the underlying reasoning behind the model's predictions. The attention mechanism has been
successfully applied in various domains. Song et al. (2017) proposed an end-to-end spatio-temporal
attention model for recognizing human actions from skeleton data, selectively attending to
distinguishable joints within each frame of the input, and assigning different levels of attention to





the output of different frames. Zhang et al. (2021) constructed an anomaly structure by incorporating
spatial attention and channel attention modules, which facilitated the creation of feature spaces
characterized by high compactness within the same class and separation between different classes,
resulting in the accurate classification of floral images. As for hydrological forecasting, Wang et al.
(2023) introduced an improved spatio-temporal attention mechanism model (STA-LSTM) for
predicting river water levels. By visualizing attention weights, they discovered that the hydrological
station closer to the outlet had greater influence, while the temporal weights decreased with
increasing historical moments. However, it should be noted that the discussed model (STA-LSTM)
considers only a single historical water level as input, neglecting the potential influence of other
relevant input features on the final prediction. This limitation underscores the need for further
research and development to explore the incorporation of multiple input features in attention
mechanisms for more comprehensive and accurate models.

Second, the DL models lack physical mechanisms. DL models primarily focus on establishing

a mapping relationship between inputs and outputs, overlooking the underlying physical
connections between them (Jiang et al., 2020). Consequently, the prediction results obtained from
DL models may be physically inconsistent or unreliable due to extrapolation or observation bias
(Reichstein et al., 2019). To address this limitation, researchers have proposed incorporating
physical constraints into the loss function, which serves as the optimization objective of DL models.
By adding physical theory as a priori knowledge, the models can be constrained to generate outputs
that are consistent with the underlying physical principles, thereby enhancing their physical
consistency. Several studies have explored this approach in different contexts. Read et al. (2019)
chose the law of energy conservation as a physical constraint in temperature simulation to build a
lake water temperature prediction model that conforms to physical theory. Wang et al. (2020)
proposed a theory-guided neural network (TgNN) framework for groundwater flow that
incorporates control equations, boundary conditions, initial conditions, and expert knowledge as
additional terms in the loss function to guide the training process. Xie et al. (2021) considered
extreme storm events, long-duration rainless events, and rainfall-runoff monotonic relationships in
the rainfall-runoff process at a daily scale and constrained LSTM with these three physical
mechanisms to improve the physical interpretability.

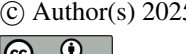



Moreover, the current inputs for the DL models in flood forecasting are mainly historical runoff,
rainfall, and evapotranspiration (Leedal et al., 2013; Rahimzad et al., 2021; Wan et al., 2019), but
the initial soil moisture is also a crucial parameter, particularly for arid watersheds (Grillakis et al.,
2016). The initial soil moisture directly affects the soil infiltration capacity, water input and output
from the soil, and ultimately, the flooding process. Therefore, the paper also explores the effect of
initial soil moisture on flood forecasting through the attention weight visualization matrix.
Based on the above research, this paper proposes a combined feature-time multi-head attention
mechanism and physical constraints model for flood forecasting, named PHY-FTMA-LSTM. The
main contributions of this work are outlined as follows: (1) The initial soil moisture in the watershed
is introduced as an input, alongside historical runoff, rainfall, and evapotranspiration, these four
input features are considered to investigate their influence on the flooding process. (2) The dual
attention module of features and time and multiple attention heads are used. The resulting attention
weight matrix is visualized to enhance the interpretability of the model, providing insights into the
importance of different features and time dynamics. (3) The physical constraints of flood forecasting
are combined with the DL models at hourly scales to enhance the physical consistency of the model.
By optimizing the loss function, the model incorporates the monotonic relationship between rainfall,
evapotranspiration, initial soil moisture, and runoff during the flooding process. This integration
ensures that the output aligns with physical laws.
The novelty of this study is that, for the first time, the attention mechanism and physical
constraints are simultaneously incorporated into the DL model based on the hourly scale, and the
important parameter of soil moisture content is added as input to forecast flood with a lead time of
1~6h in Luan River Basin in China as an example, which improves the prediction performance of
flood forecasting models while enhancing interpretability and physical law consistency. The
proposed PHY-FTMA-LSTM can effectively leverage key input information and produce prediction
results that conform to the monotonicity constraints on the water balance.
**2. Methods**
To increase the interpretability and physical consistency of DL models in flood forecasting,
this paper establishes a PHY-FTMA-LSTM model that combines the feature-time-based multi-head
attention mechanism with physical constraints (Fig. 1(a)). The attention mechanism consists of a





dual module: feature-based attention and time-based attention. In the feature-based attention module,
the model generates a feature-based attention matrix that assigns different weights to the input
features based on their importance. Similarly, the time-based attention module generates a time-
based attention matrix that assigns different weights to historical moments. By taking the dot product
of these two matrices, the model generates the feature-time-based attention matrix (Fig. 1(b)). To
enhance the modeling capability, the multi-head attention mechanism is utilized. Multiple attention
heads are computed in parallel, and their outputs are averaged to balance the influence of each
subhead. The attention weight matrix is then multiplied with the input matrix, resulting in the output
of the feature-time-based multi-head attention layer (Fig. 1 (c)). In addition, the physical constraints
of the hydrological cycle process are added to the loss function to make the output conform to the
physical laws. And the model is compared with the original LSTM, the feature-time-based attention
LSTM (FTA-LSTM), and the feature-time-based multi-head attention LSTM (FTMA-LSTM).
2.1. Long short-term memory neural network(LSTM)
The LSTM model aims to alleviate the weaknesses of ordinary RNNs in handling long-time
dynamics (Zhao et al., 2017). Different from the circular structure of the RNN hidden layer, the
hidden layer of the LSTM introduces the memory cell, which consists of an input gate, forget gate,
and output gate to selectively remember and forget the input data, and its structure is shown in Fig.
1(d). The inputs at time $t$ include the input information $x_t$ at $t$, the hidden layer state $h_{t-1}$, and the cell
state $c_{t-1}$ at $t$-1. First, the forget gate determines the extent to which cell state $c_{t-1}$ is discarded. Next,
the input gate decides how much of the current external information $x_t$ to retain and generates the
candidate cell state $\overline{c_t}$. Then, $c_t$ is updated based on the results of the forget and input gate. Finally,
the output gate decides which state features of $c_t$ are output and generates the hidden layer state
variable $h_t$ (Duan et al., 1992). The above process can be expressed as follows:
$$f_t = \sigma\left(W_f \cdot [h_{t-1}, x_t] + b_f\right) \tag{1}$$

$$i_t = \sigma\left(W_i \cdot [h_{t-1}, x_t] + b_i\right) \tag{2}$$

$$\overline{c_t} = \tanh\left(W_c \cdot [h_{t-1}, x_t] + b_c\right) \tag{3}$$

$$c_t = c_{t-1} \odot f_t + \overline{c_t} \odot i_t \tag{4}$$






$$o_t = \sigma\left(W_o \cdot [h_{t-1}, x_t] + b_o\right) \qquad (5)$$


$$h_t = \sigma \odot \tanh(c_t) \qquad (6)$$

where $W_f$, $W_i$, $W_c$, $W_o$ are the weight vectors of the three gates and the gating unit, respectively.
Similarly, $b_f$, $b_i$, $b_c$, $b_o$ are the bias vectors. $\sigma$ is the Sigmoid activation function. tanh is the hyperbolic
tangent activation function. $\odot$ denotes the vector element product.
2.2. Attention mechanism
The attention mechanism is inspired by the concept of human visual selective attention, which
helps neural networks focus on important information while disregarding irrelevant details, thereby
establishing connections between inputs and outputs (Brauwers & Frasincar, 2023; Niu et al., 2021).
By incorporating the attention mechanism, the model can allocate varying degrees of attention to
different historical moments or feature vectors within the input sequence. This enables the model to
automatically identify and prioritize the most relevant input information, leading to more accurate
modeling of flood causes and trends. Ultimately, this improves the accuracy of flood prediction
results and enhances the interpretability of the model.
In this study, a soft attention module is introduced before the original LSTM's input. This
module calculates attention weight matrices separately for input features and historical moments
and then combines them to produce a feature-time attention weight matrix.
The feature-based attention module can focus on the effects of different features on predicted
floods and improve the model's attention to important features. In this paper, the input features are
runoff, rainfall, evapotranspiration, and initial soil moisture. Let the input be a two-dimensional
matrix $X \in R^{k \times n}$, where k and n denote the number of input features and the number of historical
moments, respectively, then the input matrix at time t can be regarded as n k-dimensional vectors
$X_t = [x_1^t, x_2^t, ...., x_k^t]_{1 \times k}^T$. The input features at each time step are normalized using the softmax function
(Eq. (7) and Eq. (8)). The attention weight matrix based on the input features is obtained by
synthesizing the feature weights of all historical moments.

$$\alpha_i^t = softmax(x_i^t) = \frac{e^{-x_i^t}}{\sum_{i=1}^{k} e^{-x_i^t}} \qquad (7)$$





$$\alpha_t = \left[\alpha_1^t, \alpha_2^t, ..., \alpha_k^t\right]_{1\times k}^T \tag{8}$$

where $\alpha_i^t$ is the weight of the $i$th feature, and $\sum_{i=1}^{k}\alpha_i^t = 1$.
The time-based attention module allows simulating the relationship between different time
steps, focusing on the more important historical moments. The input matrix of features can be
viewed as $X_k = [x_k^{t-n-1}, x_k^{t-n-2}, ..., x_k^t]_{1\times n}$, and the same softmax function (Eq. (9)) is used to generate
the time-based attention weights (Eq. (10)), and the time weights of all features are synthesized to
be the attention weight matrix based on historical moments.
$$\beta_k^i = softmax(x_k^i) = \frac{e^{-x_k^i}}{\sum_{i=1}^{n}e^{-x_k^i}} \tag{9}$$

$$\beta_k = \left[\beta_1, \beta_2, ..., \beta_n\right]_{1\times n} \tag{10}$$

where $\beta_k^i$ is the weight of the $i$th time step, and $\sum_{i=1}^{k}\beta_k^i = 1$. Finally, the above two weight matrices
are multiplied element by element to obtain the attention weight matrix that focuses on both the
input features and historical moments (Eq. (11)).
$$FTA = FA \odot TA^T = \begin{bmatrix} \alpha_1^{t-n-1}\beta_1^{t-n-1} & \cdots & \alpha_1^t\beta_1^t \\ \vdots & & \vdots \\ \alpha_k^{t-n-1}\beta_k^{t-n-1} & \cdots & \alpha_k^t\beta_k^t \end{bmatrix}_{k\times n} \tag{11}$$

To enhance model expressiveness and interpretability, this study also employs a multi-head
attention mechanism. This mechanism involves passing input sequences through m independent
attention heads in parallel. Each head can be seen as a distinct representation space, enabling the
model to concurrently focus on different parts of the input. As a result, the model becomes more
capable of capturing the intricate relationships between inputs and gaining a deeper understanding
of the input data.
The multi-head attention mechanism computes $m$ sets of attention coefficients based on the
number of heads, adds the output tensor of the attention heads using the Add function, and then
balances the effects of different sub-heads by averaging operations. Finally, the average output
tensor is multiplied by the input to get the final output, which makes the attention head weights more
discriminative and better captures the relationship between sequences. The feature-time-based



multi-head attention weight matrix is as follows:
$$FTMA = \frac{1}{M}\begin{bmatrix} \sum_{m=1}^{M}\alpha_1^{t-n-1}\beta_1^{t-n-1} & \cdots & \sum_{m=1}^{M}\alpha_1^{t}\beta_1^{t} \\ \vdots & & \vdots \\ \sum_{m=1}^{M}\alpha_k^{t-n-1}\beta_k^{t-n-1} & \cdots & \sum_{m=1}^{M}\alpha_k^{t}\beta_k^{t} \end{bmatrix}_{k\times n} \quad (12)$$
where $M$ represents the number of attention heads.
2.3. Physical constraints
The LSTM is a black-box model that ignores complex physical processes, making it difficult
to maintain consistency with the basic principles of flood forecasting (Yokoo et al., 2022). To
overcome this limitation, the physical constraints can be combined with the DL models to enhance
the physical consistency by modifying the model loss function and transforming the prior
knowledge of flood forecasting into the penalty term of the loss function. A soft penalty is often
utilized to enforce constraints on the model's behavior (Karniadakis et al., 2021), ensuring
adherence to physical principles such as conservation and monotonicity.
In the DL models for flood forecasting, the occurrence of flooding due to heavy rainfall is
influenced by various factors, including rainfall intensity, evapotranspiration, infiltration, and
storage dynamics. When considering the input features of rainfall, evapotranspiration, and initial
soil moisture, it is important to maintain a monotonic relationship between each feature and the
resulting runoff. However, the traditional DL models disregard the physical relationships between
inputs and outputs. This lack of consistency with the physical principles of water balance equations
undermines the overall reliability of the model. Therefore, this study incorporates inequality
constraints to enforce the desired monotonic relationships between rainfall, evapotranspiration,
initial soil moisture, and runoff. Under the assumption that all other input variables remain
unchanged, a new time series of rainfall, evapotranspiration, and initial soil moisture is generated
respectively by applying a small random increase using the random.uniform function. These new
time series are then combined with the unchanged time series to form new input data. The difference
between the predicted values corresponding to the new data and the predicted values corresponding
to the original input data is calculated. This difference is then converted into a specific loss value
using the ReLU function and added to the loss function.





**Fig. 1.** (a) The PHY-FTMA-LSTM model architecture. (b) Feature-time-based attention matrix

generation process for each attention head. (c) Feature-time-based multi-head attention workflow.
(d) The internals of LSTM cells.
For rainfall, the runoff should increase if there is a slight increase in rainfall at the current time
step, provided that other variables are constant, and the monotonic relationship and losses for
rainfall-runoff are expressed as follows:
$$f\left[p(t)+\Delta p,t\right]-f\left[p(t),t\right]\geq 0 \tag{13}$$

$$Loss_p=\frac{1}{N_p}\sum\nolimits_{i=1}^{N_p}\left\{\mathrm{ReLU}\left\{f\left[p(t),t\right]-f\left[p(t)+\Delta p,t\right]\right\}\geq 0\right\}^2 \tag{14}$$

where $\Delta p$ is the small increase in rainfall, $Loss_p$ is the error in the monotonic relationship of rainfall
runoff, $N_p$ is the sample length of the perturbed rainfall, and ReLU is the response function.
For evapotranspiration, the runoff should decrease if there is a slight increase in
evapotranspiration at the current time step, provided that other variables are constant, and the
monotonic relationship and losses for evapotranspiration runoff are expressed as follows:
$$f\left[e(t)+\Delta e,t\right]-f\left[e(t),t\right]\leq 0 \tag{15}$$

$$Loss_e=\frac{1}{N_e}\sum\nolimits_{i=1}^{N_e}\left\{\mathrm{ReLU}\left\{f\left[e(t),t\right]-f\left[e(t)+\Delta e,t\right]\right\}\leq 0\right\}^2 \tag{16}$$

where $\Delta e$ is the small increase in evapotranspiration, $Loss_e$ is the error in the monotonic relationship
of evapotranspiration runoff, $N_e$ is the sample length of the perturbed evapotranspiration.
For soil moisture, the runoff should increase if the initial soil moisture of the watershed
increases slightly for each flood event, provided that other variables are constant, and the monotonic
relationship and losses between initial soil moisture and runoff are expressed as follows:
$$f\left[s(t)+\Delta s,t\right]-f\left[s(t),t\right]\geq 0 \tag{17}$$

$$Loss_s=\frac{1}{N_s}\sum\nolimits_{i=1}^{N_s}\left\{\mathrm{ReLU}\left\{f\left[s(t),t\right]-f\left[s(t)+\Delta s,t\right]\right\}\geq 0\right\}^2 \tag{18}$$

where $\Delta s$ is the small increase in initial soil moisture, $Loss_s$ is the error in the monotonic relationship
of initial soil moisture runoff, $N_s$ is the sample length of the perturbed initial soil moisture.
Based on the above physical constraints of flood forecasting, the loss function of the traditional
LSTM model is improved with the following equation:
$$Loss=\lambda_{data}Loss_{data}+\lambda_p Loss_p+\lambda_e Loss_e+\lambda_s Loss_s \tag{19}$$





where *Loss* is the loss function of the LSTM guided by the physical constraints of flood forecasting;
$Loss_{data}$ is the mean square error of the observed and predicted values of the LSTM; $\lambda_{data}$、$\lambda_p$、$\lambda_e$、
$\lambda_s$ are the weighting coefficients of different losses, respectively. To treat the three physical
constraints equally, the weighting coefficients of the four losses are set to {0.7, 0.1, 0.1, 0.1}.
2.4. Evaluation metrics
To evaluate the accuracy of different models for flood forecasting, the Nash-Sutcliffe efficiency
(NSE), Kling–Gupta efficiency (KGE), the coefficient of determination ($R^2$), root mean square error
(RMSE), and mean absolute error (MAE) are selected for evaluation. The specific equations are as
follows:
$$NSE = 1 - \frac{\sum_{i=1}^{n}\left(Q_t - Q_t'\right)^2}{\sum_{i=1}^{n}\left(Q_t - \overline{Q_t}\right)^2} \tag{20}$$

$$KGE = 1 - \sqrt{\left(R-1\right)^2 + \left(\alpha-1\right)^2 + \left(\beta-1\right)^2} \tag{21}$$

$$R^2 = \frac{\left(\sum_{i=1}^{n}\left(Q_t - \overline{Q_t}\right)\left(Q_t' - \overline{Q_t'}\right)\right)^2}{\sum_{i=1}^{n}\left(Q_t - \overline{Q_t}\right)^2 \sum_{i=1}^{n}\left(Q_t' - \overline{Q_t'}\right)^2} \tag{22}$$

$$RMSE = \sqrt{\frac{\sum_{i=1}^{n}\left(Q_t - Q_t'\right)^2}{n}} \tag{23}$$

$$MAE = \frac{1}{n}\sum_{i=1}^{n}\left|Q_t - Q_t'\right| \tag{24}$$

where $Q_t$ is the observed value; $Q_t'$ is the predicted value; $\overline{Q_t}$ is the observed mean value; $\overline{Q_t}'$ is
the mean value of the predicted series; $\alpha$ between the standard deviation of the predicted value and
that of the observed value; $\beta$ is the ratio between the mean of the predicted value and that of the
observed value; $n$ is the total number of samples. The NSE is commonly used to evaluate
hydrological prediction models, KGE considers the contribution of mean, variance and correlation
on model performance, $R^2$ is often used to evaluate the linear correlation between the forecast
process and the observed process. The values of NSE, KGE and $R^2$ range from 0 to 1. The closer the
result is to 1, the more accurate the forecast result is and the higher the model credibility is. RMSE
and MAE are used to reflect the degree of deviation between the predicted and observed values, the





smaller the value the smaller the deviation.

## 3. Study area and data

3.1. Study area

In this study, the watershed controlled by the Sandaohezi station in the Luan River Basin was selected as the study area. The Luan River originates from the northern foot of Bayangurtu Mountain in Hebei Province, with a total length of 888 km, and flows through Inner Mongolia, Hebei, and Liaoning provinces before injecting into the Bohai Sea at Laoting County, Hebei Province. The station is in the middle reaches of the mainstream of the Luan River, controlling a watershed area of 17100 km$^2$, accounting for about 40% of the total area of the Luan River basin. Geographically, it is located between 115.5°E to 117.7°E longitude and 40.7°N to 42.7°N latitude. The elevation of the study area ranges from 370 to 2300 m, with a high northwest to low southeast topography. Except for the upstream origin of the dam plateau, the rest of the area is dominated by mountainous terrain. The northwest of the basin is located in the temperate continental climate zone, precipitation is scarce and concentrated in summer; the southeast is located in the temperate monsoon climate zone, with cold, dry winters and hot, rainy summers. The average annual temperature of the basin ranges from 5 to 12°C, and the average annual runoff is about 480 million m$^3$. The average annual rainfall is about 500mm, and the spatial and temporal distribution of rainfall within the year is uneven, mainly concentrated from May to September, and the precipitation decreases from south to north. Floods in the basin are mostly formed by heavy rainfall, which is short-lived and strong, making the flooding process steep up and steep down, often causing disasters in the downstream areas. Consequently, accurate flood forecasting is of utmost importance for effective flood control and water resources management in the Luann River basin. The location of the study area and the stations are shown in Fig. 2.

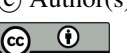

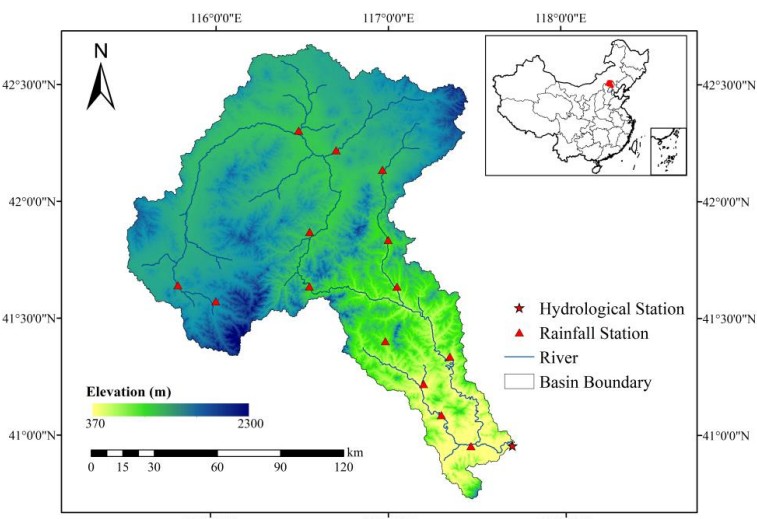


**Fig.2.** Geographical location of the study area and hydrological and rainfall stations.
3.2. Data
The rainfall and runoff data were obtained from the Hydrological Yearbook of the Haihe River
Basin, including rainfall data from 15 rainfall stations, such as Sandaohezi, Zhangbaiwan, and
Baorono, and runoff data from Sandaohezi hydrological station. The period covers 39 years from
1964 to 1989, 1991, and 2006 to 2017. There is a gap in the data for 1990 and 1992 to 2005 due to
incomplete data collection.
The evapotranspiration and soil moisture data were obtained from the Global Land Surface
Data Assimilation System (GLDAS) using the GLDAS-Noah model product 0.25°×0.25° spatial
resolution, 3h temporal resolution dataset, and the evapotranspiration data were averaged backward
3h, and the soil moisture data were instantaneous values. Among them, GLDAS-2.0 provides data
from 1964 to 2014, and GLDAS-2.1 provides data from 2015.
In this study, 30 flood events during the 39 years were selected (Table 1), and the collected
observed runoff data were linearly interpolated to 1h step data, the observed rainfall data were
averaged to 1h step data, and the Tyson polygon method was used to derive the areal rainfall. For
evapotranspiration and soil moisture, the average values were calculated for each grid in the
watershed at each period, where the soil moisture was taken as the initial soil moisture before the
onset of rainfall for each flood event. Twenty flood events were used for model training, ten flood





events were used for model validation.
Since different input features have different magnitudes, maximum-minimum normalization
was used to process the input data into the range [0,1], see Eq. (25).
$$x_{norm} = \frac{x_i - x_{min}}{x_{max} - x_{min}} \tag{25}$$

where $x_{norm}$ is the normalized data, $x_i$ is the original data, and $x_{min}$ and $x_{max}$ are respectively the
minimum and maximum values of the original data.
**Table 1** Flood events used in the study.

| Dataset | Flood number | Peak discharge (m³/s) | Year | Duration (month/day/hour) |
|---------|--------------|------------------------|------|----------------------------|
|  | 1 | 314.2 | 1964 | 08/01/04-08/09/12 |
|  | 2 | 218 | 1964 | 08/13/02-08/16/00 |
|  | 3 | 313 | 1965 | 07/17/20-07/21/12 |
|  | 4 | 204 | 1966 | 07/27/16-07/31/20 |
|  | 5 | 260 | 1968 | 07/27/12-07/30/22 |
|  | 6 | 154 | 1969 | 08/20/12-08/27/12 |
|  | 7 | 296 | 1971 | 07/17/15-07/29/08 |
|  | 8 | 153 | 1972 | 07/19/08-07/24/08 |
|  | 9 | 742 | 1973 | 08/12/04-08/26/08 |
|  | 10 | 213 | 1975 | 08/11/00-08/16/08 |
| Training | 11 | 218 | 1978 | 08/25/12-09/03/08 |
|  | 12 | 246 | 1982 | 07/22/12-07/29/16 |
|  | 13 | 313 | 1983 | 08/04/00-08/11/20 |
|  | 14 | 400 | 1985 | 08/24/05-08/31/04 |
|  | 15 | 210 | 1986 | 08/08/04-08/13/08 |
|  | 16 | 87.5 | 1987 | 08/19/12-08/23/04 |
|  | 17 | 465 | 1991 | 06/10/04-06/18/00 |
|  | 18 | 70.1 | 2008 | 08/10/00-08/16/00 |
|  | 19 | 149 | 2010 | 07/30/17-08/04/20 |
|  | 20 | 80.4 | 2015 | 07/27/16-07/31/16 |
|  | 21 | 241 | 1965 | 08/26/21-08/30/20 |
|  | 22 | 260 | 1967 | 06/27/12-06/29/22 |
|  | 23 | 164 | 1970 | 07/14/12-07/16/04 |
|  | 24 | 506.7 | 1974 | 07/23/12-08/06/08 |
| Validation | 25 | 313 | 1979 | 08/13/04-08/21/08 |
|  | 26 | 132 | 1985 | 08/11/16-08/14/04 |
|  | 27 | 212 | 1989 | 06/03/22-06/07/04 |
|  | 28 | 205 | 2011 | 08/14/10-08/20/04 |





| | | | |
|---|---|---|---|
| 29 | 95.9 | 2013 | 07/21/08-07/25/16 |
| 30 | 84.2 | 2013 | 08/13/09-08/21/00 |

3.3. Model construction
This study is based on Python 3.9, and the Numpy, Pandas, and Scikit-Learn packages in
Python are used for data processing, and the LSTM, FTA-LSTM, FTMA-LSTM, and PHY-FTMA-
LSTM models are constructed using the Keras library in TensorFlow.
The model inputs are runoff, rainfall, evapotranspiration, and initial soil moisture for a
specified time step, and the outputs are the discharge from 1 to 6h of the lead time. All four models
use the ReLU activation function, which avoids gradient vanishing and is more effective compared
to the tanh and sigmoid functions. The Adam optimizer is used and the LSTM layer is a single layer,
with the number of attention heads set to 3 for the FTMA-LSTM and PHY-FTMA-LSTM. The mean
square error is the loss function of the four models, and for PHY-FTMA-LSTM it incorporates
physical constraints, as shown in Eq. (19). To avoid overfitting, all models use the early stopping
and set the maximum number of epochs to 200.
To construct the base models, the common values of the DL model parameters are used as the
initial values. The base models have an observed input time step of 12 hours, a learning rate of 0.001,
batch size of 64, and hidden units set to 128. After evaluating the performance of the base models,
parameter optimization is performed separately for each of the four models, considering that the
optimal parameter combinations may differ among the models. The goal is to study the effects of
the input time step and three hyperparameters (learning rate, batch size, and hidden units) on the
model performance. The ranges used for parameter optimization are as follows: input time step of
3 to 24 hours, learning rate of 0.00001 to 0.01, batch size of 16 to 256, and hidden units of 32 to
512. A single parameter is varied while the other parameters are taken as their initial values.
Considering the stochastic nature of the DL model running process, each of the four models is
repeated five times for each lead time, and the results with the best prediction performance are
selected for analysis.
**4. Results**
4.1. Model optimization
The LSTM, FTA-LSTM, FTMA-LSTM, and PHY-FTMA-LSTM base models are established





individually, and their average NSE values during the 1-6 hour lead time, measure to evaluate flood
prediction accuracy, are found to be 0.925, 0.930, 0.936, and 0.950, respectively. These results
indicate that all four base models can effectively predict flooding events. In order to determine the
optimal parameter combination for each model and how individual parameter variations affect the
model performance, the following parameters are investigated while keeping the other three
parameters constant: input time step, learning rate, batch size, and hidden units.

Regarding the input time step of observations, experiments are conducted by varying the time

step within a certain range. The result depicted in Figure 3(a) shows that the average NSE value for
all four models is highest at a time step of 12 hours and decreases with increasing time step. The
worst performance is observed at a time step of 24 hours. This observation suggests that longer input
sequences introduce more noise, and the inclusion of extraneous information adversely affects the
final prediction. Therefore, a 12-hour input time step is identified as the optimal choice for flood
forecasting in all four models and is adopted for subsequent experiments.

For the learning rate, tests are performed using a learning rate ranging from 0.00001 to 0.01.

The finding, presented in Figure 3(b), indicates that the performance of the four models is
comparable at learning rates of 0.01 and 0.001. However, when the learning rate is set to 0.0001 and
0.00001, the models exhibit slow convergence and degrade performance rapidly. Considering the
possibility of failure to converge at a very high learning rate, a combined analysis suggests a learning
rate of 0.001 as the optimal choice for all four models in the subsequent studies.

The batch size optimization ranges from 16 to 256. The result depicted in Figure 3(c)

demonstrates varying performances of the four models with different batch sizes. The LSTM model
achieves the highest average NSE of 0.932 at a batch size of 128. Similarly, the FTA-LSTM model
attained its highest average NSE of 0.932 at a batch size of 32. On the other hand, the FTMA-LSTM
and PHY-FTMA-LSTM models reach their highest average NSE values at a batch size of 64, with
0.936 and 0.950, respectively. Consequently, the optimal batch size for flood forecasting is
determined as 128, 32, 64, and 64 for the LSTM, FTA-LSTM, FTMA-LSTM, and PHY-FTMA-
LSTM models, respectively. These batch sizes are employed for subsequent studies.

Regarding the hidden units, tests are conducted with the count varying from 32 to 512. Figure

3(d) illustrates the distinct performances of the four models concerning different hidden units. The





LSTM model achieves the highest average NSE of 0.925 with 64 hidden units. The FTA-LSTM and
FTMA-LSTM models attain their highest average NSE values of 0.935 and 0.939 with 256 hidden
units, respectively. In contrast, the PHY-FTMA-LSTM model reaches the highest average NSE of
0.950 at 128. Accordingly, the optimal hidden units for flood prediction are identified as 64, 256,
256, and 128 for the LSTM, FTA-LSTM, FTMA-LSTM, and PHY-FTMA-LSTM models,
respectively.
Considering the above parameter optimization process, the model parameters used in the
subsequent study are as follows (Table 2). Notably, the PHY-FTMA-LSTM model consistently
outperforms the other three models across various parameter values, exhibiting the smallest
variation in NSE. These findings indicate that the PHY-FTMA-LSTM model proposed in this paper
offers the best and most stable performance.
**Table 2** Parameters of models.

| Models | Input time step | Learning rate | Batch size | Hidden units |
|---|---|---|---|---|
| LSTM | 12 | 0.001 | 128 | 64 |
| FTA-LSTM | 12 | 0.001 | 32 | 256 |
| FTMA-LSTM | 12 | 0.001 | 64 | 256 |
| PHY-FTMA-LSTM | 12 | 0.001 | 64 | 128 |


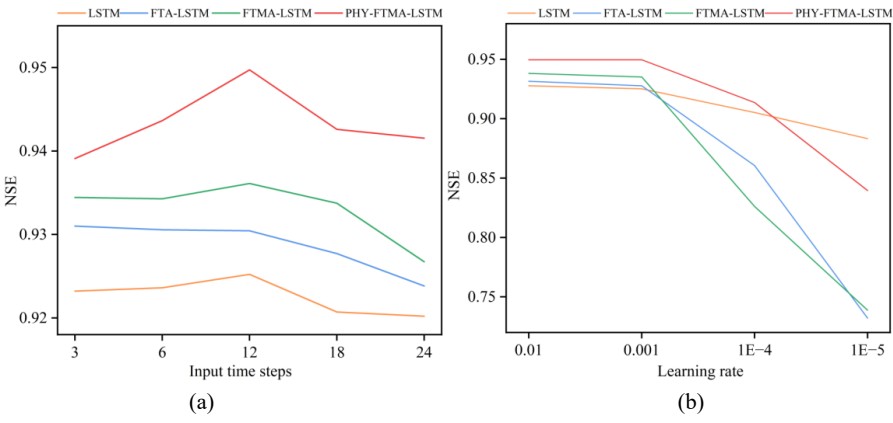

(a)                                    (b)



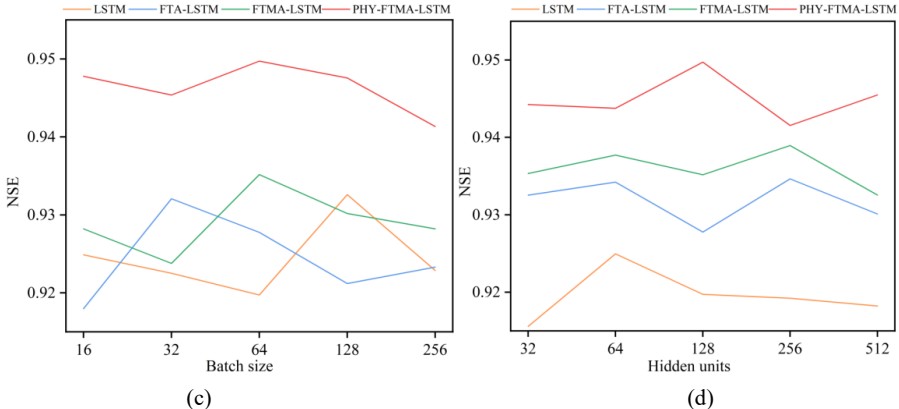

(c)                                              (d)

**Fig.3.** The NSE values for 6 lead times with different (a) input time steps of observations, (b) learning rate, (c) batch size, and (d) hidden units.

4.2. Model performance evaluation

The LSTM, FTA-LSTM, FTMA-LSTM, and PHY-FTMA-LSTM models are constructed using the optimal parameters mentioned above, the evaluation metrics of the forecasting performance of the four models in the training and validation stages are shown in Table 3 and Table 4. All the metrics of the four models almost outperform the validation period in the training period. And with the increase of the lead time, the gap between the performance of the models in the training period and the testing period gradually increases. It can be seen that the three models based on the attention mechanism outperform the original LSTM model in all lead times. It indicates that the dual attention module of time and feature proposed in this paper effectively focuses on the more significant historical moments and feature variables, improving the performance of the LSTM model. Among the attention-based models, the FTMA-LSTM model, which utilizes a multi-headed attention mechanism, achieves better performance than the FTA-LSTM model with a single attention head in most cases. This demonstrates that the parallel computation of the multi-head attention mechanism enables the model to emphasize more important information in the input compared to the single-head attention mechanism. Furthermore, the PHY-FTMA-LSTM model, which incorporates physical constraints, outperforms the other three models across almost all metrics. Specifically, at the lead time t+1, compared to the original LSTM model, the PHY-FTMA-LSTM model shows an improvement in NSE, KGE, and $R^2$, increasing from 0.977 to 0.988, from 0.953 to 0.984 and from 0.979 to 0.988, respectively. Additionally, the RMSE and MAE decrease by 27.4% and 49.6%,





respectively. At the lead time t+6, NSE increases from 0.865 to 0.908, KGE from 0.851 to 0.905,
$R^2$ from 0.886 to 0.911, and RMSE and MAE decrease by 21.1% and 15.1%, respectively. These
results mean that incorporating physical constraints enables the DL model to understand the
monotonic relationship presented in the flooding process, improving forecast accuracy by enhancing
the model's physical consistency.
As the lead time increases, the performance of all four models declines, suggesting that their
robustness and generalization gradually deteriorate. However, the extent of the decline in the four
model metrics varies. In terms of NSE, when transitioning from a 1-hour to a 6-hour lead time, the
PHY-FTMA-LSTM model exhibits the smallest decline of 0.065 during the training period, while
the LSTM, FTA-LSTM, and FTMA-LSTM models experience decreases of 0.072, 0.079, and 0.073
respectively. During the validation period, the NSE value decreases by 0.080 for the PHY-FTMA-
LSTM model and by 0.112, 0.109, and 0.104 for the LSTM, FTA-LSTM and FTMA-LSTM models,
respectively. Maintaining high accuracy in longer lead times is crucial in practical applications.
Extended lead times necessitate more comprehensive information for accurate predictions,
presenting challenges for the models. Nonetheless, the PHY-FTMA-LSTM model exhibits minimal
degradation, indicating its superior ability to adapt to longer lead times and maintain high precision.
This superiority may be attributed to the unique characteristics and structure of the PHY-FTMA-
LSTM model. It likely encompasses considerations of physical factors and key input features,
enabling a better capture of flood complexity and variability. This advantage positions the model
favorably in scenarios requiring predictions further into the future.

**Table 3** Performance of the four models for flood forecasting at different lead times for training.

| Lead times/h | Models | NSE | KGE | $R^2$ | RMSE | MAE |
|---|---|---|---|---|---|---|
| | LSTM | 0.977 | 0.964 | 0.980 | 16.14 | 7.14 |
| t+1 | FTA-LSTM | 0.986 | 0.972 | 0.987 | 12.32 | 5.19 |
| | FTMA-LSTM | 0.990 | 0.977 | 0.990 | 10.62 | 4.76 |
| | PHY-FTMA-LSTM | **0.992** | **0.984** | **0.992** | **9.65** | **4.03** |
| | LSTM | 0.959 | 0.944 | 0.963 | 21.52 | 11.29 |
| t+2 | FTA-LSTM | 0.966 | **0.983** | 0.967 | 20.93 | **7.85** |
| | FTMA-LSTM | 0.969 | 0.960 | 0.972 | 18.54 | 8.80 |
| | PHY-FTMA-LSTM | **0.976** | 0.949 | **0.977** | **16.56** | 9.10 |



| Lead times/h | Models | NSE | KGE | $R^2$ | RMSE | MAE |
|---|---|---|---|---|---|---|
| | LSTM | 0.943 | 0.945 | 0.948 | 25.09 | 13.91 |
| t+3 | FTA-LSTM | 0.949 | 0.943 | 0.952 | 22.05 | 11.02 |
| | FTMA-LSTM | 0.954 | **0.963** | 0.955 | 21.14 | **10.79** |
| | PHY-FTMA-LSTM | **0.958** | 0.955 | **0.963** | **20.01** | 11.45 |
| | LSTM | 0.933 | 0.915 | 0.942 | 27.59 | 15.83 |
| t+4 | FTA-LSTM | 0.945 | **0.956** | 0.948 | 23.06 | 14.57 |
| | FTMA-LSTM | 0.948 | 0.953 | 0.949 | **22.12** | **13.75** |
| | PHY-FTMA-LSTM | **0.950** | 0.948 | **0.955** | 23.63 | 14.27 |
| | LSTM | 0.929 | 0.917 | 0.929 | 29.16 | 18.91 |
| t+5 | FTA-LSTM | 0.930 | **0.942** | 0.931 | 27.99 | 16.37 |
| | FTMA-LSTM | 0.934 | 0.925 | 0.937 | 26.08 | 16.18 |
| | PHY-FTMA-LSTM | **0.937** | 0.931 | **0.937** | **25.58** | **15.19** |
| | LSTM | 0.905 | 0.900 | 0.917 | 33.29 | 19.78 |
| t+6 | FTA-LSTM | 0.907 | 0.913 | 0.913 | 33.63 | 17.86 |
| | FTMA-LSTM | 0.917 | 0.926 | 0.919 | 30.59 | **15.83** |
| | PHY-FTMA-LSTM | **0.927** | **0.949** | **0.929** | **28.05** | 16.04 |

Figure 4 displays the scatter plots for the LSTM, FTA-LSTM, FTMA-LSTM, and PHY-
FTMA-LSTM models during the training and validation periods. When the foresight period is 1
hour, all models demonstrate predictions that closely track the ideal 1:1 line. The PHY-FTMA-
LSTM model outperforms the others, exhibiting the narrowest scatter distribution. However, as the
lead time increases, the scatter plots of the four models show varying degrees of deterioration,
becoming more uneven and scattered. The high discharge prediction error increases in the training
period, and the validation period reveals numerous underestimated discharges. Among them, the
PHY-FTMA-LSTM model performs the best (with the narrowest scatter distribution), followed by
the FTA-LSTM and FTMA-LSTM models. The LSTM model performs the worst. Notably, during
the validation period, for longer foresight periods, the high flow scatter of all models deviates further
from the ideal 1:1 line. One possible explanation is the scarcity of high flow instances in the training
data. As the lead time increases, the models struggle to capture the necessary information, leading
to underestimation and poorer predictions. For a foresight period of 6 hours, the scatter plots of the
LSTM, FTA-LSTM, and FTMA-LSTM models both in the training and validation periods exhibit
discrete distributions. In contrast, the PHY-FTMA-LSTM model's scatter plot shows the narrowest
band and is closest to the ideal 1:1 line. Consequently, the PHY-FTMA-LSTM model achieves the





highest prediction accuracy, effectively reducing prediction errors for longer lead times. The FTA-
LSTM and FTMA-LSTM models follow while the LSTM model performs the worst in terms of
prediction accuracy.
**Table 4** Performance of the four models for flood forecasting at different lead times for validation.

| Lead times/h | Models | NSE | KGE | $R^2$ | RMSE | MAE |
|---|---|---|---|---|---|---|
| t+1 | LSTM | 0.977 | 0.953 | 0.979 | 15.84 | 8.45 |
| | FTA-LSTM | 0.985 | 0.969 | 0.985 | 12.65 | 6.28 |
| | FTMA-LSTM | 0.987 | 0.975 | 0.988 | 11.83 | 5.04 |
| | PHY-FTMA-LSTM | **0.988** | **0.984** | **0.988** | **11.50** | **4.26** |
| t+2 | LSTM | 0.956 | 0.939 | 0.961 | 21.83 | 11.94 |
| | FTA-LSTM | 0.961 | **0.974** | 0.961 | 19.07 | 10.22 |
| | FTMA-LSTM | 0.967 | 0.950 | 0.970 | 18.83 | 9.52 |
| | PHY-FTMA-LSTM | **0.968** | 0.954 | **0.970** | **18.56** | **9.45** |
| t+3 | LSTM | 0.934 | 0.928 | 0.938 | 27.09 | 14.93 |
| | FTA-LSTM | 0.942 | 0.927 | 0.943 | 25.07 | 13.49 |
| | FTMA-LSTM | 0.948 | **0.947** | 0.951 | 23.66 | **12.56** |
| | PHY-FTMA-LSTM | **0.952** | 0.945 | **0.955** | **21.57** | 12.74 |
| t+4 | LSTM | 0.918 | 0.914 | 0.929 | 28.15 | 16.43 |
| | FTA-LSTM | 0.928 | 0.938 | 0.933 | 28.17 | **14.20** |
| | FTMA-LSTM | 0.931 | **0.946** | 0.933 | 28.44 | 16.24 |
| | PHY-FTMA-LSTM | **0.939** | 0.938 | **0.944** | **26.13** | 14.59 |
| t+5 | LSTM | 0.898 | 0.890 | 0.900 | 36.43 | 22.83 |
| | FTA-LSTM | 0.905 | 0.911 | 0.910 | 32.54 | 19.36 |
| | FTMA-LSTM | 0.915 | 0.915 | **0.920** | 30.43 | 20.52 |
| | PHY-FTMA-LSTM | **0.918** | **0.930** | 0.919 | **30.33** | **16.65** |
| t+6 | LSTM | 0.865 | 0.851 | 0.886 | 40.61 | 23.77 |
| | FTA-LSTM | 0.876 | 0.894 | 0.886 | 37.38 | 20.57 |
| | FTMA-LSTM | 0.883 | 0.889 | 0.896 | 36.52 | 20.65 |
| | PHY-FTMA-LSTM | **0.908** | **0.905** | **0.911** | **32.02** | **20.18** |


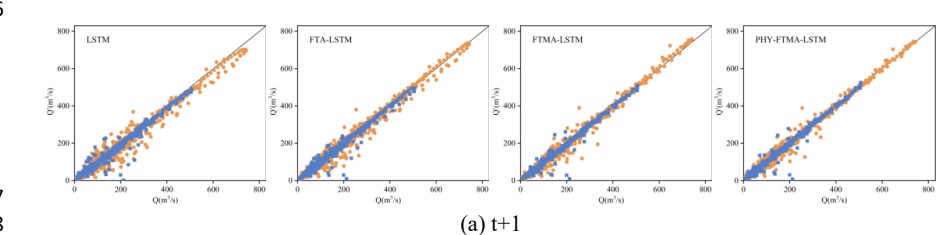

(a) t+1



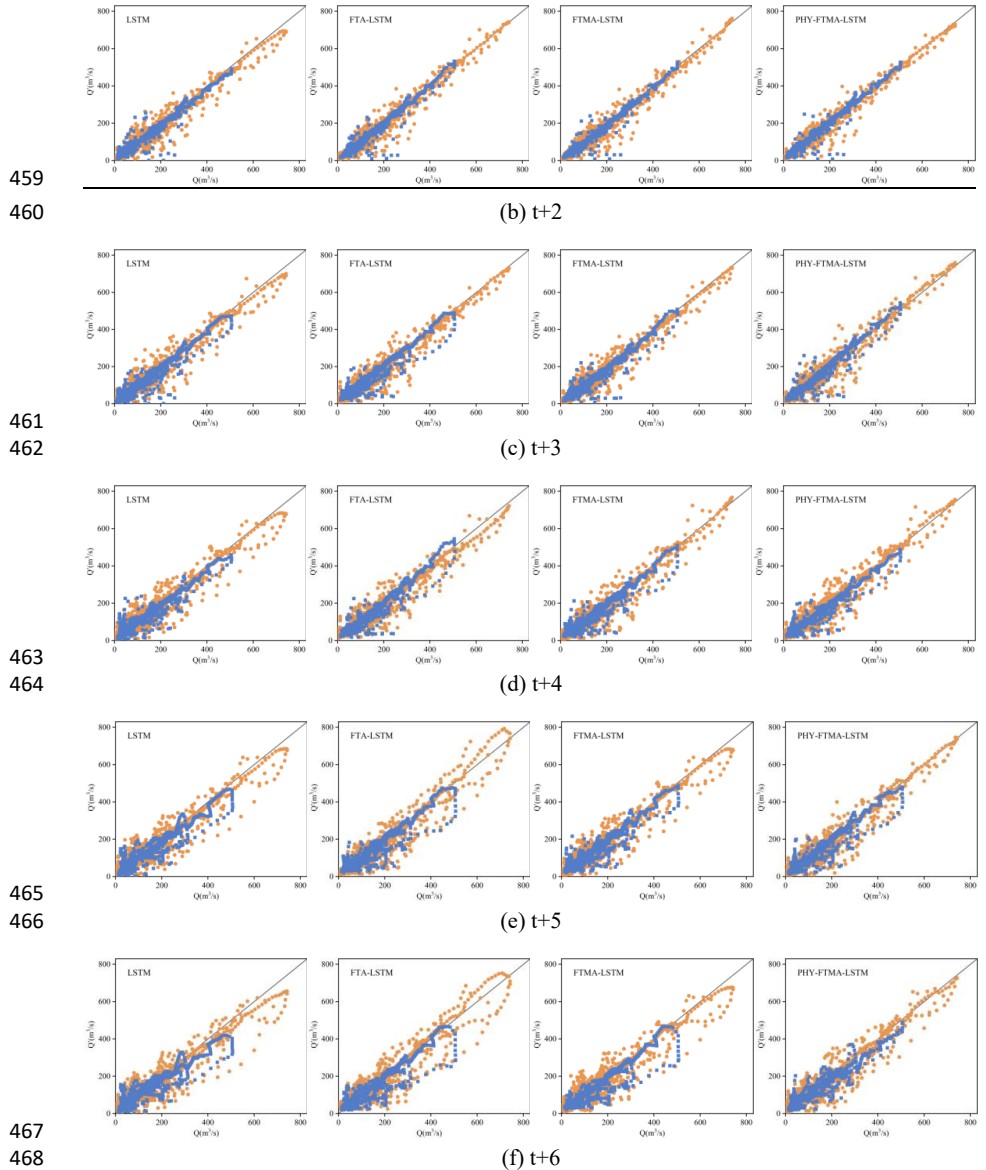


(b) t+2

(c) t+3

(d) t+4

(e) t+5

(f) t+6
**Fig.4.** Scatter plots of observed and predicted discharges in the training and validation stages, in
which yellow represents the training stage and blue represents the validation stage.
4.3. Typical flood event forecast results

Floods in the basin are mainly two types, single-peak and double-peak, so two typical flood

events were selected to analyze the specific flood process: a double-peak flood event (19740723)
with a peak discharge of 507 $m^3$/s and 290 $m^3$/s, and a single-peak flood event (19790813) with a



peak discharge of 313 m³/s. Fig. 5 and Fig. 6 illustrate the flood processes of the two events predicted
by the four models. It can be observed that as the lead time increases, the prediction hydrographs
from all four models gradually deviate from the observed values and the three evaluation metrics
decrease. Notably, the LSTM model exhibits the greatest decline in prediction performance,
followed by the FTA-LSTM and FTMA-LSTM models. In contrast, the PHY-FTMA-LSTM model
demonstrates relatively better performance across the evaluated flood events.

Based on the analysis of prediction hydrographs, the four models exhibit better performance in

predicting the double-peak flood event compared to the single-peak flood event. Additionally, the
models demonstrate higher accuracy in predicting the rising stage of floods in contrast to the falling
stage. Specifically, the prediction errors increase as the duration of the flood increases, and there is
a time lag in predicting the occurrence of the second flood peak. When it comes to the single-peak
flood event, the predictions by the four models display greater fluctuations, and the time lag problem
is more pronounced, along with an overestimation of the peak discharge.

Regarding the 19740723 flood event, the LSTM model generally underestimates the discharge

values, and the discrepancy with the observed hydrograph gradually increases as the lead time
increases. Although the FTA-LSTM and FTMA-LSTM models also underestimate the discharge,
their errors are reduced, indicating improved performance compared to the LSTM model. In contrast,
the PHY-FTMA-LSTM model predicts the flood hydrograph more accurately. However, when the
foresight period is 6 h, the PHY-FTMA-LSTM model experiences significant prediction errors due
to anomalous fluctuations.

For the 19790813 flood event, the LSTM model demonstrates a noticeable deviation from the

predicted hydrograph with increasing lead times. The FTA-LSTM and FTMA-LSTM models
exhibit better performance, as their predicted hydrographs are closer to the observed ones. However,
there is some overestimation of the peak discharge in these models. Additionally, all three models
suffer from a more severe time lag issue in longer foresight periods. In contrast, the PHY-FTMA-
LSTM model shows smaller volume errors and is closer to the observed hydrograph. Nevertheless,
this model exhibits a more pronounced overestimation of the peak discharge.

In conclusion, the LSTM model exhibits poor prediction performance for longer lead times.

On the other hand, the FTA-LSTM, FTMA-LSTM, and PHY-FTMA-LSTM models show improved





performance with longer lead times and higher forecasting accuracy. Among these models, the PHY-
FTMA-LSTM model stands out by producing better predictions for both single-peak and multi-peak
flood events, but it may encounter challenges with predicting anomalous fluctuations at longer lead
times. Additionally, the PHY-FTMA-LSTM model mitigates the issue of time lag to some extent by
considering the physical monotonicity relationship.

(a) t+1                                                   (b) t+2


(c) t+3                                                   (d) t+4




(e) t+5               (f) t+6

**Fig.5.** Comparison of observed and predicted values of the 19740723 flood event by the four
models.

(a) t+1               (b) t+2


(c) t+3               (d) t+4


(e) t+5               (f) t+6

**Fig.6.** Comparison of observed and predicted values of the 19790813 flood event by the four



models.
4.4. Visual attention analysis

To investigate the changes in features and time attention of PHY-FTMA-LSTM with different

lead times, the attention weights of PHY-FTMA-LSTM are visualized in Fig. 7. The figure consists
of six subplots representing lead times ranging from t+1 to t+6.

From Fig. 7, it can be observed that the distribution pattern of the weights remains relatively

similar across different forecasting periods. The temporal attention weights decrease as the historical
moment increases. Among the feature-based weights, runoff has the highest proportion, followed
by rainfall, and finally the initial soil moisture and evapotranspiration. These results align with
hydrological principles, where runoff is considered the most direct manifestation of the flooding
process and holds the highest importance. Rainfall, as the main driver of flood formation,
significantly influences flooding. In contrast, the effects of initial soil moisture and
evapotranspiration in the basin are more indirect and therefore receive lower weights. In the case of
the Luan River basin, which is relatively arid, the initial soil moisture of the basin is typically not
saturated. During a rainfall-induced flood, there is a possibility of transitioning from infiltration-
excess runoff to saturation-excess runoff. Hence, special attention should be given to the role of the
initial soil moisture, which carries slightly greater relative importance than evapotranspiration.

As the forecasting horizon extends, the feature-time-based weights of the model become more

concentrated, with the time-based weights gradually moving forward. Consequently, the model
places more emphasis on the values that are closer to the current moment. Additionally, the feature-
based attention module exhibits a gradual increase in attention to rainfall while decreasing attention
to evapotranspiration and the initial soil moisture. Notably, runoff retains its status as the most
influential factor.

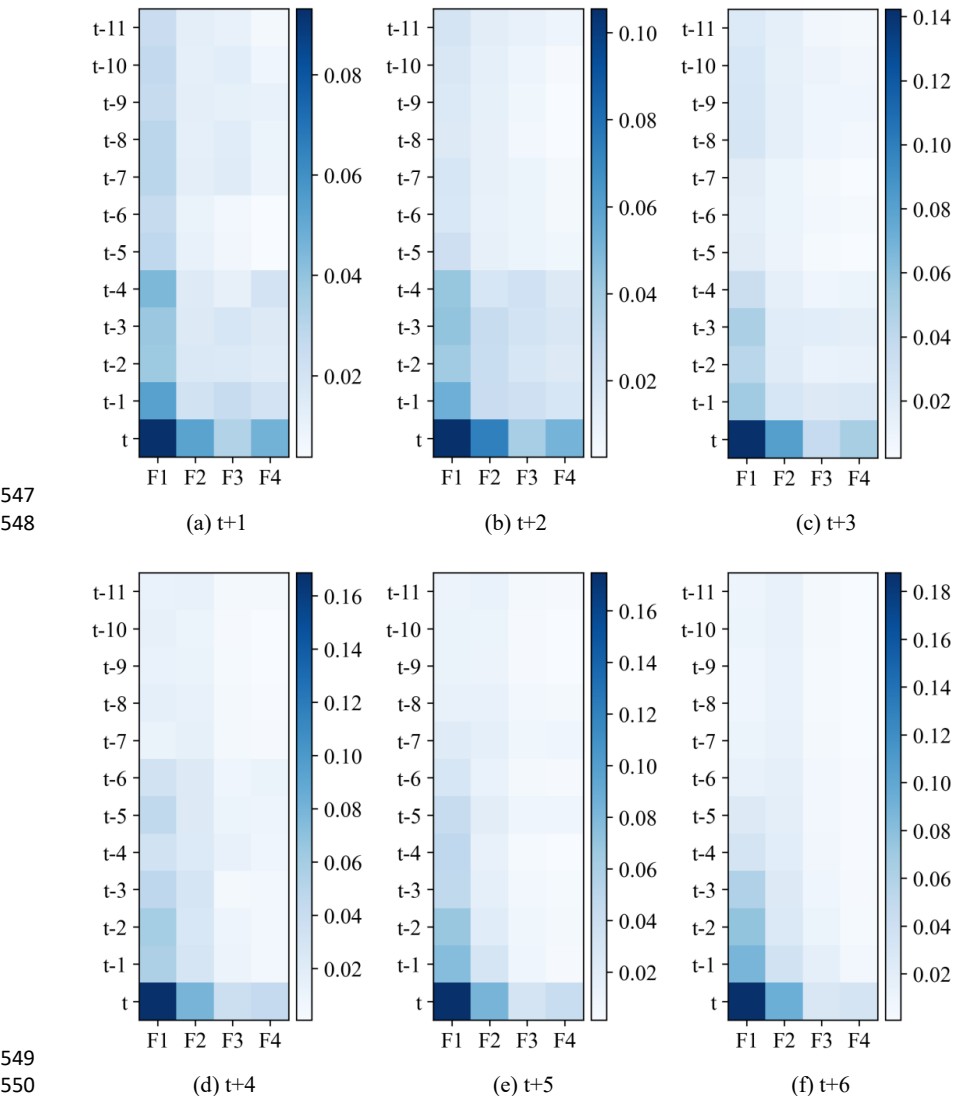

(a) t+1          (b) t+2          (c) t+3

(d) t+4          (e) t+5          (f) t+6

**Fig.7.** The visualization of feature-time-based attention weights of the PHY-FTMA-LSTM. The
X-coordinate variables F1 to F4 represent the input features of runoff, rainfall, evapotranspiration,
and initial soil moisture of the watershed, respectively. The Y-coordinate variables represent the
input history moments.
## 5. Discussion
The input time step of observations, learning rate, batch size, and hidden units are significant
parameters that influence the performance of the model, and the optimal parameters may vary for





different structural models (Xiang et al., 2020; Cao et al., 2022). In this study, four models, namely
LSTM, FTA-LSTM, FTMA-LSTM, and PHY-FTMA-LSTM, have been constructed. To ensure that
each model achieves its optimal prediction performance and to investigate the impact of different
parameter variations on model performance, the same parameter values are utilized to build the four
base models individually. After confirming that the base models meet the accuracy requirements for
flood forecasting, the optimal parameter combination for each model is determined. This is done by
selecting the parameter value associated with the highest NSE obtained through single parameter
tuning. The single parameters are changed while keeping the initial values of the other three
parameters constant. This approach ensures that the subsequent analysis reflects the best
performance achievable by each model's specific structure. Moreover, it enables a more explicit
evaluation of the performance changes resulting from the addition of attention mechanisms and
physical constraints to the model.

570   In terms of model performance evaluation metrics, the PHY-FTMA-LSTM model

demonstrates the best overall performance. However, a closer examination reveals that its KGE
score may not necessarily be optimal. This could be attributed to the comprehensiveness of the KGE
metric, which considers factors such as correlation, mean consistency, and variance consistency of
the flow. Fluctuations in the KGE score may arise from various uncertainties related to data quality,
model structure, and flood forecasting.

576   With an increase in the forecast period, the performance of the model, particularly the LSTM

model, shows a significant decrease, consistent with the findings reported by Xu et al. (2021). They
provided NSE, RMSE, and Bias indices for the LSTM model in forecast periods of 1~12 hours,
demonstrating that the LSTM model meets prediction requirements for short forecast periods.
However, as the forecast period extends, the accuracy diminishes, leading to underestimation of
flood peaks and significant fluctuations. Similar conclusions were drawn in the studies conducted
(Cui et al., 2021; Ding et al., 2020). The longer the foresight period, the lower the correlation
between input and output variables. The models face increased difficulty due to the lack of future
information and the challenges associated with flood forecasting.

585   The addition of an attention mechanism effectively enhances the accuracy of flood forecasting

in the original LSTM model. As the lead time increases, the temporal weights gradually shift



forward, causing the model to pay greater attention to values closer to the current moment. This
finding aligns with the conclusions of studies on temporal attention conducted by Ding et al. (2020)
and Wang et al. (2023). However, there is a difference between their studies and the current one, as
they incorporated a spatial attention module to focus on the relevance of spatial locations, while this
study introduces a feature attention module to highlight the importance of different input features in
flood forecasting.
Incorporating physical constraints into the model enhances the understanding of the monotonic
relationships between variables in the flooding process and improves the physical consistency of
the model. This study considers the monotonic relationships between precipitation, evaporation,
initial soil moisture content, and runoff in the watershed. In a study by Xie et al. (2021), three
physical conditions related to the rainfall-runoff forecasting process were encoded into the loss
function at the daily scale. Experimental results on 531 watersheds in the CAMELS dataset showed
that the model achieved an improvement from 0.52 to 0.61 in the NSE mean compared to the LSTM
model. In this study, flood forecasting is performed at a finer time scale, specifically at the hourly
scale, and additional monotonic relationship constraints between evapotranspiration, initial soil
water content, and runoff are incorporated.
Flood forecasting is challenged by various complex factors such as meteorological conditions
and rainfall patterns, and the uncertainty of these factors increases over time (Cheng et al., 2021;
Hu et al., 2019). Consequently, the model is prone to significant prediction errors. When the forecast
period extends to 6 hours, each model exhibits a significant deviation from the observed hydrograph
and more anomalous fluctuations. In this study, the maximum prediction period of the model is set
at 6 hours, and the effects of longer prediction periods need further investigation. In future research,
we propose exploring additional methods to address these limitations and enhance the performance
of our model. One potential avenue is the incorporation of error correction methods such as K
nearest neighbor (KNN) and backpropagation (BP) algorithms. Additionally, data assimilation
techniques, such as ensemble Kalman filter and particle filter methods, can be used to assimilate the
latest observed data and improve real-time forecasting accuracy. These approaches have the
potential to extend the forecasting period of flood prediction.





## 6. Conclusions

This research introduces a DL model called PHY-FTMA-LSTM, which combines feature-time-based multi-head attention mechanisms with physical constraints. The primary aim is to explore how incorporating interpretability and physical constraints into DL models affects flood forecasting accuracy. The evaluation of the flood forecasting results from 1 to 6 h during the foresight period in the Luan River basin yields the following conclusions:

(1) The attention mechanism that considers both features and time effectively enhances the model's prediction performance, surpassing that of the original LSTM model. The FTMA-LSTM model, equipped with an increased number of attention heads, further improves accuracy by considering more information through parallel computation. Taking the integration of physical constraints into account, the PHY-FTMA-LSTM model achieves the best performance, exhibiting stable results. For a lead time of t+1, the NSE, KGE, $R^2$, RMSE, and MAE reaches 0.988, 0.984, 0.988, 11.50, and 4.26, respectively. Additionally, NSE, KGE, and $R^2$ also could reach 0.908, 0.905, and 0.911 for a lead time of t+6.

(2) The incorporation of a feature-time-based multi-head attention mechanism improves interpretability by directing attention to the most valuable features and historical moments within the inputs. The weight matrix visualization reveals that runoff emerges as the most influential feature in flood forecasting, followed by rainfall, and finally initial soil moisture and evapotranspiration. Furthermore, the weight distribution becomes more concentrated with increasing lead time.

(3) The model combines physical constraints by considering the monotonic relationships between rainfall, evapotranspiration, initial soil moisture, and runoff at an hourly scale. This augmentation significantly improves the model's predictive capacity for flood processes, including flood peaks, while reducing the lag time.

In this study, we have successfully incorporated both the attention mechanism and physical mechanism into a DL model to improve the accuracy of flood prediction while ensuring interpretability and physical consistency. In future research, we recognize that there is room for further enhancing the interpretability of our model. We suggest exploring alternative interpretation techniques to gain deeper insights into the model's decision-making process. Furthermore, the combination of physical mechanisms and DL models can be expanded by incorporating more



detailed basin subsurface information and exploring different integration methods that consider both
physical mechanisms and DL models.

## Code and data availability

The rainfall and flood data and model codes used in this study could be available online
(https://github.com/zran1/PHY_FTMA_LSTM.git). The evapotranspiration and initial soil moisture
data are extracted from GLDAS Noah Land Surface Model (Beaudoing et al., 2019; D. Beaudoing
et al., 2020), which is freely available at https://disc.gsfc.nasa.gov/datasets.

## Author contributions

Ting Zhang: Conceptualization, Methodology, Writing-original draft, Writing-review &
editing. Ran Zhang: Conceptualization, Methodology, Software, Validation, Writing-original draft.
Jianzhu Li: Validation, Writing-review & editing. Ping Feng: Validation, Writing-review & editing.

## Competing interests

The contact author has declared that none of the authors has any competing interests.

## Acknowledgements

This work was supported by the National Key Research and Development Program of China
(2023YFC3006501, 2023YFC3006503), National Natural Science Foundation of China (No.

52279022, 52079086).

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
