# Peer review of "Deep learning of flood forecasting by considering"

_Hydrology and Earth System Sciences, 2024_

## Author Response (AR1)

**Response to Reviewers**

Deep forecasting learning flood considering of by

interpretability and physical constraints

Ting Zhang \*, Ran Zhang, Jianzhu Li, Ping Feng

State Key Laboratory of Hydraulic Engineering Intelligent Construction and Operation, Tianjin

University, Tianjin 300072, China

Corresponding author: Ting Zhang (zhangting hydro@tju.edu.cn)

Dear editor and reviewers,

Thank you very much for giving us the opportunity to review the manuscript and make

valuable comments on our research, the comments and suggestions made on our manuscript by the

two reviewers were encouraging and helpful. We have addressed all these major points and other

issues carefully and revised the manuscript accordingly. As per the editor's suggestions, we have

removed the China map portion from Figure 2 to avoid potential controversies. This revision

retains the latitude/longitude markers that sufficiently identify our study area. Meanwhile, the

abstract has been revised to enhance its conciseness and accessibility for general readers.

Additionally, we have included an additional funding source in the Acknowledgements section.

We have provided detailed, point-by-point responses to the reviewers' comments in the following

pages. Note that the reviewers' comments are presented in italics, and our responses are in Times

New Roman and blue font. In addition, all the line numbers in the responses refer to the revised

manuscript. All changes made to the manuscript are marked in red font. Please do not hesitate to

| contact us if you have any quest | ions or require | e any additional | information. | Thank | you for | your |
|----------------------------------|-----------------|------------------|--------------|-------|---------|------|
| consideration.                   |                 |                  |              |       |         |      |
|                                  |                 |                  |              |       |         |      |
| Sincerely,                       |                 |                  |              |       |         |      |
| Ting Zhang                       |                 |                  |              |       |         |      |

To the Reviewer #1's comments, we make the following responses and changes in the manuscript:

**1. Comment: Abstract**

It is recommended to avoid using unexplained acronyms, as this may hinder comprehension for the reader. Including a brief environmental and geological context of the studied basin would help justify the choice of the forecast horizon t + 6. This parameter is highly dependent on the characteristics of the river considered and may be excessive or not meaningful in other fluvial contexts. Consequently, the results cannot be generalized without appropriate caution.

Response: Thank you for your insightful comment. All acronyms have been explicitly defined upon their first appearance in the revised manuscript, including LSTM, PHY-FTMA-LSTM and NSE. And we have modified the abstract to make it more concise and reader-friendly.

**In the revised manuscript, Page 1, Line 8-18:**

Deep learning models show promise for flood forecasting but often lack interpretability and physical realism. To bridge this gap, we enhance traditional Long Short-Term Memory (LSTM) networks by integrating: (1) a feature-time attention mechanism that emphasizes critical input features and historical moments by learning dynamic weights, and (2) physics-guided constraints that enforce fundamental hydrological principles by considering the monotonic relationships between inputs and outputs. Tested in China's Luan River Basin for 1-6 hour flood predictions, the proposed physics-guided feature-time-based multi-head attention mechanism LSTM (PHY-FTMA-LSTM) outperforms standard LSTM and attention-only variants. It achieves exceptional accuracy with Nash-Sutcliffe efficiency (NSE) values of 0.988 at t+1 and maintains strong performance at 0.908 at t+6, offering valuable insights for enhancing interpretability and

physical consistency in deep learning approaches.

The selection of the forecast period was determined through a comprehensive evaluation of multiple factors, including basin-specific geological and environmental characteristics (added in Section 3.1 Line 288-297). Statistical analysis indicated that the flood concentration time in the study basin typically ranges between 6-12 hours. Meanwhile, we referred to the 6-hour forecasting horizon following Wang et al. (2023), whose methodology demonstrated successful water-level forecasting in the Han River Basin (covering>30,000 km²). Furthermore, extending the prediction horizon was constrained by the inherent black-box nature of deep learning models, which exhibited significant performance degradation over longer periods.

**Manuscript Reference:**

Wang, Y., Huang, Y., Xiao, M., Zhou, S., Xiong, B., Jin, Z., 2023. Medium-long-term prediction of water level based on an improved spatio-temporal attention mechanism for long short-term memory networks. J. Hydrol. 618(129163).

**In the revised manuscript, Page 12, Line 288-297:**

Based on geological conditions and geomorphological features, the area can be divided into two dominant landform types: plateau and mountainous terrain. The plateau dominates the northern part of the basin, with elevations ranging from 1400 to 1600 meters and a gentle channel gradient averaging approximately 0.5‰. The remaining area comprises mountainous terrain, exhibiting complex topography shaped by prolonged denudation and erosion. This zone features steep mountains, densely distributed hills, and interspersed basins, with slope angles varying between 20° and 40°. In certain areas, rivers demonstrate intense downward cutting action, resulting in

significantly steeper channel gradients — typically 2–6‰, while some medium and small tributaries exceed 20‰. Notably, flood wave propagation velocities reach 2.0-3.5 m/s due to these topographic conditions.

**2. Comment: Study area and data**

It is unclear why the experiment was conducted in this particular watershed. What are its characteristics? Why is it relevant? How does it differ from others? Will the results obtained be valid only for this site, or are they generalizable? This should be clarified.

Response: Thank you for your insightful comment. The Luan River Basin is a large watershed (44880 km²) spanning Hebei, Inner Mongolia, and Liaoning provinces, holding critical geopolitical and economic significance. It serves as a vital ecological conservation zone and water source for the Beijing-Tianjin-Hebei region. However, as a seasonal river, it alternates between rapid flood peaks during rainy seasons and frequent dry-season flow interruptions. Channel encroachment and increased agricultural/industrial water withdrawals have exacerbated downstream flow breaks, diminishing flood conveyance capacity and heightening disaster risks. Its complexity and representativeness establish it as a paradigm for flood forecasting research in similar basins.

Our modeling methods are universally applicable, while the parameter configurations exhibit distinctiveness that necessitates calibration based on local catchment characteristics for implementation in alternative basins.

3. Comment: Lines 297–306: The data sampling frequency is not specified, even though it is a

fundamental parameter.

Response: Thank you for your insightful comment. The sampling frequency in our study complies

with China's National Hydrological Data Compilation Standards, which require dynamic rather

than fixed-interval sampling. For flood hydrographs, additional 2-3 measurements are taken

before the rising limb and after recession stabilization to facilitate baseflow separation, with

critical points captured during rising/falling limb transitions and peak periods (minimum 3-5

measurements around peak discharge). Precipitation monitoring also adopts intensified logging

frequency during heavy rainfall events to ensure data accuracy under extreme conditions. This

adaptive sampling protocol ensures comprehensive hydrological process documentation while

meeting technical requirements for flood forecasting analysis. So the data were processed as 1h

time step according to Line 321-326.

4. Comment: Figure 2: it is recommended to change the colors, as the triangles and the star are

not clearly visible.

Response: Thank you for your insightful comment. We have made the hydrological and rainfall

stations clearer in the revised manuscript by changing the colors and sizes.

In the revised manuscript, Page 13, Fig.2:

Fig.2. Geographical location of the study area and hydrological and rainfall stations.

5. Comment: The dataset split into training and validation sets appears to be the main critical issue of the study. What rule was followed? Currently, the most accepted strategy is to divide the dataset into three parts (training, validation, and test), using the validation set during batch steps. Why wasn't this approach followed? Is it due to the limited number of available cases? An explanation is required.

What happens if the events that compose the three subsets are changed? Does the predictive performance of the models vary? Using techniques like cross-validation or bootstrapping would allow for the analysis of error distributions. How stable is a model trained multiple times on the same initial dataset? Answering these questions would strengthen the scientific approach of the paper, moving it beyond a simple application. The results presented seem fragile as they might depend on the initial, arbitrary assignment of events to the training, validation, and test phases.

Response: Thank you for highlighting datasets concerns. Our data partitioning strategy was

rigorously based on two primary considerations: First, with only 30 historical floods, and most of them had been short-lived, resulting in a limited sample size, dividing the three datasets would leave  $\leq 6$  events to test, which was insufficient to capture spatiotemporal heterogeneity. We will also add more floods in the future if they become available. Second, this approach followed established precedents by researchers including Lv et al., Read et al., Xie et al., and Jiang et al., who used dual dataset partitioning and not only for flood forecasting.

As can be seen from Table 1, our dataset partitioning had taken into consideration key flood characteristics including temporal occurrence, peak discharge, and flood duration. This intentional design ensured that both datasets comprehensively covered a wide spectrum of flood features, thereby enabling thorough training and rigorous testing of the model. Furthermore, we had experimented with exchanging some floods within the training and validation sets. While preserving the diversity profiles of flood characteristics in both datasets, the model prediction performance changed little.

Importantly, we implemented a sliding window mechanism with a 12-timestep window length and 1-timestep stride for sample construction. This configuration ensured:

- (a) Continuous temporal coverage by advancing the window progressively at each computational step.
  - (b) Maximized data utilization through 92% overlap between consecutive windows.
  - (c) Effective capture of hydrological process evolution patterns characteristic of flood events.

While cross-validation wasn't explicitly used, sliding windows inherently achieved dynamic CV (used by Gao et al., Kao et al. And Ding et al. for flood forecasting). Each timestep participated in multiple windows, akin to data recycling in CV.

**Manuscript Reference:**

Lv, N., Liang, X., Chen, C., Zhou, Y., Li, J., Wei, H., Wang, H., 2020. A long Short-Term memory cyclic model with mutual information for hydrology forecasting: A Case study in the xixian basin.

Adv. Water Resour.

Read, J. S., Jia, X., Willard, J., Appling, A. P., Zwart, J. A., Oliver, S. K., Karpatne, A., Hansen, G. J. A., Hanson, P. C., Watkins, W., Steinbach, M., Kumar, V., 2019. Process-Guided Deep Learning Predictions of Lake Water Temperature. Water Resour. Res. 55(11), 9173-9190.

Xie, K., Liu, P., Zhang, J., Han, D., Wang, G., Shen, C., 2021. Physics-guided deep learning for rainfall-runoff modeling by considering extreme events and monotonic relationships. J. Hydrol. 603, 127043.

Jiang, S., Zheng, Y., Solomatine, D., 2020. Improving AI System Awareness of Geoscience Knowledge: Symbiotic Integration of Physical Approaches and Deep Learning. Geophys. Res. Lett. 47(e2020GL08822913).

Gao et al., 2020. Short-term runoff prediction with GRU and LSTM networks without requiring time step optimization during sample generation. J. Hydrol., 589 (2020), Article 125188.

Kao et al., 2020. Exploring a long short-term memory based encoder-decoder framework for multi-step-ahead flood forecasting.J. Hydrol., 583 (2020), Article 124631.

Ding, Y., Zhu, Y., Feng, J., Zhang, P., Cheng, Z., 2020. Interpretable spatio-temporal attention LSTM model for flood forecasting. Neurocomputing. 403, 348-359.

6. Comment: Are 30 events sufficient to train deep learning models? The size of the original dataset and the derived datasets is not clear. I suggest conducting a distributional analysis of the

events. If the analysis focuses on a limited number of cases, they should be hydrologically analyzed and shown to be statistically representative of the hydrology of the basin under study.

Response: Thank you for highlighting datasets concerns. The hydrological records for the Luan River Basin are inherently limited, with available data spanning discontinuous periods (1964-1989, 1991, and 2006-2017), amounting to 39 years of flood records. When selecting specific floods, we had checked the rainfall runoff data of each flood in order to ensure the reliability and representativeness of the hydrological data, and finally selected 30 floods under the premise of guaranteeing the inclusion of three types of flood magnitudes, namely large, medium and small, and covering the single-peak and multi-peak flooding processes. In addition, the sliding window mechanism (12-timestep window, stride=1) generated 4025 temporally correlated training samples from the 30 events, effectively. We believe that the number of flood events and samples could basically reflect the watershed situation and support the model training.

7. Comment: Line 323: Indicate the version of the TensorFlow library used.

Response: Thank you for your insightful comment. We used TensorFlow 2.9.1, which we have added in the revised manuscript.

In the revised manuscript, Page 15, Line 337:

.....are constructed using the Keras library in TensorFlow 2.9.1.

8. Comment: Line 327: Provide a citation for the activation functions employed.

Response: Thank you for your insightful comment. We have added reference to the activation functions used while providing the formal definition of Rectified Linear Unit (ReLU).

**In the revised manuscript, Page 15, Line 340:**

All four models use the Rectified Linear Unit (ReLU) activation function (Nair & Hinton, 2010).

**Manuscript Reference:**

Nair, V., Hinton, G. E., 2010. Rectified linear units improve restricted boltzmann machines vinod nair. Omnipress.

9. Comment: Line 330: it is unclear how overfitting is being mitigated by early stopping. It must be demonstrated that the models are not affected by overfitting. Furthermore, splitting the dataset into three sets is a fundamental first step to prevent both overfitting and underfitting.

Response: Thank you for your insightful comment. We employed early stopping to monitor changes in the loss function (Mean Squared Error, MSE) and terminated process if the loss showed no improvement for 20 consecutive epochs. We have also added this to the revised manuscript.

Additionally, although not mentioned in the article, during each run we visualized the loss curve to observe its trend, thereby assessing model convergence and potential overfitting. Overfitting typically manifests as strong performance on the training set but poor generalization on the validation set. However, as demonstrated by the metrics we provided, the model exhibited good generalization capability on the validation set. Regarding the dataset, as previously explained, the limited number of flood events constrained the dataset splitting.

**In the revised manuscript, Page 15, Line 345-347:**

To avoid overfitting, all models employ early stopping based on the mean squared error (MSE) loss function, with a maximum iteration limit of 200 epochs. The training process automatically

terminates if no improvement in loss is observed for 20 consecutive epochs.

**10. Comment: Results**

Tables 3 and 4: It is advisable to replace the tables (which can be included as supplementary material to ensure transparency of the raw data) with plots showing the metrics as a function of lead time for each model. This would help reveal potential trends and the presence or absence of overfitting. Additionally, the reported results may lack statistical validity and could be coincidental. It is necessary to repeat the training procedures, as mentioned above, to assess the robustness of the outcomes.

What if the error metrics were computed only for data exceeding a certain threshold (statistical or physical)? Focusing on peak flood events, would the metrics change? Would more patterns emerge?

Response: Thank you for your insightful comment. We have replaced the tables with plots showing the metrics as a function of lead time for each model (Fig.4 and Fig.5) and added the tables to the supplementary material (Table S1 and S2). As detailed in Line 357-359, all four models were repeated for five runs at each lead time to assess stability. It is confirmed that there was little difference in performance between runs, and the best performing implementation was selected for final analysis to ensure that it would be ready for use in the event of a flood forecast.

In addition, we initially employed boxplots of peak discharge relative errors but peer reviewers noted both the absence of significant inter-model differences and insufficient representation of process dynamics, such as rising/falling limb error. Thus, we transitioned to observed vs. predicted

scatterplots (Fig.6), which could reveal full-process error patterns, identify extreme-event outliers

**In the revised manuscript, Page 18, Line 415-416:**

the evaluation metrics of the forecasting performance of the four models in the training and validation periods are shown in Figure 4 and Figure 5. Detailed metrics data can be found in the supplementary material (Table S1 and S2).

**In the revised manuscript, Page 20-21, Line 452-467:**

Fig.4. Performance of the four models for flood forecasting at different lead times for training (a)

NSE, (b) KGE, (c) R2, (d) RMSE and (e) MAE.

**Fig.5.** Performance of the four models for flood forecasting at different lead times for validation (a) NSE, (b) KGE, (c) R2, (d) RMSE and (e) MAE.

**11. Comment: Figure 4: The axis labels are not legible.**

Response: Thank you for your insightful comment. We have revised the font size to ensure that the labeling in Figure 6 (formerly Figure 4) are clear and appropriately sized.

**In the revised manuscript, Page 22-23, Line 487-500:**

**Fig.6.** Scatter plots of observed and predicted discharges in the training and validation periods, in which yellow represents the training period and blue represents the validation period.

12. Comment: This observation applies to all time horizons, but is particularly evident for t+5 and t+6: for observed discharges above approximately 300 m3/s, an anomalous behavior appears in the scatterplot points, forming a curve. In my experience, these points likely correspond to a specific event that the model fails to simulate correctly, tending to underestimate

the flows. Suppose this hypothesis is confirmed by the authors. In that case, it should be discussed, as it would reveal an interesting phenomenon: the model is unable to overestimate flow in advance and instead tends to underestimate it as lead time increases.

These models seem to suffer from a common limitation: the inability to anticipate runoff before the onset of precipitation. This limitation may be understandable given the lack of meteorological forecast input to the model. Nonetheless, this observation opens up interesting research avenues that the authors are encouraged to explore in the discussion and conclusions.

Finally, if the hypothesis that those outlier points belong to a single event holds true, the most significant errors in predicting large events should be analyzed in detail. All these aspects could serve as input for a revision of the discussion and conclusions, enhancing the scientific impact of the paper, which in its current form lacks significant novelty.

Response: Thank you for your insightful analysis of the systematic flow underestimation at high discharges (>300 m³/s). We confirm and deeply appreciate your hypothesis. The outliers indeed cluster within the 19740723 flood event (validation set peak). Our analysis have been added to the revised manuscript.

Furthermore, we acknowledge this constraint in our current modeling framework - the absence of meteorological forecast inputs restricts runoff anticipation capability prior to precipitation events.

In both the Discussion and Conclusion sections, we have added content highlighting the issues of current research lacking weather forecasting inputs and structural constraints in models.

All of the constructive critiques have profoundly shaped our research trajectory, and we thank you for elevating the practical relevance of this work.

**Discussion section**

**In the revised manuscript, Page 30, Line 636-643:**

Notably, across all forecast periods — particularly at t+5 and t+6 — scatterplot points (Fig.6.) exhibit deviant behavior forming curve patterns for discharge values exceeding approximately 300m3/s. The analysis reveals that the outliers primarily cluster during the 19740723 flood event, mainly attributable to training dataset limitations. This extreme event featured both an exceptionally prolonged duration and high peak discharge — characteristics absent from the training data. Consequently, the model demonstrates insufficient capacity to simulate such threshold-exceeding events, yielding suboptimal performance. However, as this represents an extreme scenario, model accuracy is expected to improve with expanded data accumulation.

**In the revised manuscript, Page 31, Line 648-662:**

While our framework currently caps at 6-hour predictions, extending this horizon requires confronting two fundamental constraints: (1) Input deficiency: The absence of real-time meteorological forecasts prevents runoff anticipation prior to precipitation; (2) Structural saturation: Memory decay in recurrent units limits long-range dependency capture. To address current limitations, future research will pursue a dual-track improvement strategy: Near-term efforts will focus on implementing error correction techniques, specifically K-nearest neighbors (KNN) and backpropagation (BP) algorithms, coupled with advanced data assimilation methods such as Ensemble Kalman and Particle filters to enhance real-time forecasting accuracy. While more fundamental enhancements will involve the strategic integration of numerical weather prediction inputs — specifically the European Centre for Medium-Range Weather Forecasts (ECMWF) and China Meteorological Administration Global Forecast System (CMA-GFS)

datasets — to enable pre-rainfall runoff anticipation and systematically extend predictive lead times beyond the current 6-hour threshold. Thereby addressing both immediate performance gaps and long-term capability requirements in flood forecasting.

**Conclusions section**

**In the revised manuscript, Page 32, Line 689-698:**

While our current framework demonstrates strong performance within 6-hour predictions, we recognize two key constraints for extending this horizon: the input deficiency due to missing real-time meteorological forecasts and the structural saturation caused by memory decay in recurrent units. To address these limitations, future research will provide improvements through error correction techniques and data assimilation, as well as fundamental enhancements through the integration of ECMWF/CMA-GFS numerical weather prediction inputs to enable pre-rainfall runoff prediction and extend the forecast period beyond 6 hours. Additionally, we suggest exploring other interpretation techniques to deepen understanding of the model's decision-making, while expanding the physical-DL integration through more detailed basin subsurface information and novel combination methods.

To the Reviewer #2's comments, we make the following responses and changes in the manuscript:

1. Comments: Clarification on training data size: The manuscript states that only 20 flooding

events are used for training, with each event lasting less than 10 days. Could the authors specify

the total number of training samples (e.g., input-output pairs or sequences) generated from these

events? This information is important for evaluating the robustness and generalizability of the

model.

Response: Thank you for raising this important clarification. The training samples were generated

through a sliding window approach with a 12-timestep window length and 1-timestep stride for

sample construction. This resulted in 2859 unique training samples (input-output sequence pairs).

Data size have been added to the revised manuscript.

In the revised manuscript, Page 16, Line 376-377:

The samples are constructed through a sliding window, resulting in the generation of 2859 training

samples and 1166 validation samples.

2. Comments: Physics-based loss in PHY-FTMA-LSTM (Line 224-251): Further clarification is

needed regarding the implementation of the physics-based loss. Specifically, how are the

perturbations  $\delta$  e,  $\delta$  s and  $\delta$  t, introduced during training? Are fixed values pre-specified and

added to the input variables? If so, what are the chosen values, and how are they justified?

Explicit details on this setup would greatly improve the reproducibility and interpretability of the

method.

Response: Thank you for this critical technical inquiry. We didn't introduce fixed perturbation

values. As specified in Line 213-220 of our implementation, while keeping other input variables unchanged, we employed the random uniform function (a random number generator producing values from a uniform distribution within specified bounds) to apply random minor increments within the range [0, 0.1) to the temporal sequences of precipitation, evaporation, and initial watershed soil moisture. This process generated new perturbed temporal sequences for these variables, which were then combined with the unchanged variables' sequences to form modified input datasets.

The difference between the runoff simulation values derived from these perturbed inputs and those from the original inputs was calculated and subsequently transformed into specific loss values via the ReLU function (ensuring non-negative loss). These computed loss values were then incorporated into the overall loss function for model optimization.

We've made minor changes to this section.

**In the revised manuscript, Page 8, Line 213-220:**

Under the assumption that all other input variables remain unchanged, a new time series of rainfall, evapotranspiration, and initial soil moisture is generated respectively by applying random minor increments within the range [0, 0.1) using the random.uniform function. These new time series are then combined with the unchanged time series to form new input data. The difference between the predicted values corresponding to the new data and the predicted values corresponding to the original input data is calculated. This difference is then converted into a specific loss value using the ReLU function and added to the loss function.

3. Comments: Terminology clarification (Line 120): The term "dot product" is typically reserved

for operations between vectors, whereas matrix operations such as the one described are more commonly referred to as element-wise multiplication or Hadamard product. Based on the following context, it appears that the authors intended to apply an element-wise product rather than a dot product. I recommend revising the terminology to avoid confusion and ensure mathematical accuracy.

Response: We sincerely appreciate this precise technical correction. You are absolutely correct that the operation described in Line 118 constitutes an element-wise multiplication (Hadamard product) rather than a dot product. We have revised this terminology throughout the manuscript to ensure mathematical accuracy.

**In the revised manuscript, Page 5, Line 118:**

By taking the element-wise product of these two matrices, the model generates the feature-time-based attention matrix.

4. Comments: Undefined abbreviations (Line 181): The abbreviations FA and TA are introduced without prior definition. For clarity, all abbreviations should be clearly defined at first mention to ensure readability for a broad academic audience.

Response: We sincerely appreciate this careful observation. You are absolutely correct that the abbreviations "FA" and "TA" are inadvertently undefined at first mention in Line 181-182. We have implemented the revisions to ensure clarity.

**In the revised manuscript, Page 7, Line 181-182:**

where FA represents feature-based attention weight matrix, TA represents time-based attention weight matrix.

5. Comments: Figure 1 clarity: Figure 1, particularly subplot (b), is difficult to interpret. The label "head\_m" appears to encompass multiple attention mechanisms, including feature attention, time attention, and feature-time attention—not solely multi-head attention as the label may imply. I suggest renaming the label in subplot (b) to more accurately reflect its composite structure and enhance reader comprehension.

Response: We appreciate this feedback. To enhance clarity, we have revised the label for subplot (b) to: Per-head Feature-Time Attention. This represents the formation of feature-based attention and time-based attention matrices from inputs followed by element-wise product.

In the revised manuscript, Page 9, Line 222:

**Fig. 1.** (a) The PHY-FTMA-LSTM model architecture. (b) Feature-time-based attention matrix generation process for each attention head. (c) Feature-time-based multi-head attention workflow. (d) The internals of LSTM cells.

6. Comments: Labeling in Figure 5: In Figure 5, it would be more intuitive to label the x-axis using calendar dates (e.g., MM-DD-HH) rather than elapsed time in hours. Using time in hours may be easily confused with forecast lead times, potentially causing misinterpretation. I

recommend updating the x-axis to calendar dates to improve clarity and reader understanding.

Response: Thank you for this constructive suggestion. We agree that using calendar dates on the x-axis of Figure 7 (formerly Figure 5) would enhance temporal clarity and avoid potential confusion with forecast lead times. We have updated x-axis labels to MM-DD-HH format.

**In the revised manuscript, Page 25-26, Line 540-556:**

**Fig.7.** Comparison of observed and predicted values of the 19740723 flood event by the four models.(The x-axis displays dates in MM-DD-HH format, representing month, day, and hour respectively)

**Fig.8.** Comparison of observed and predicted values of the 19790813 flood event by the four models.

---

## Author Response (AR2)

**Response to Reviewers**

considering Deep learning flood forecasting of by

interpretability and physical constraints

Ting Zhang \*, Ran Zhang, Jianzhu Li, Ping Feng

State Key Laboratory of Hydraulic Engineering Intelligent Construction and Operation, Tianjin

University, Tianjin 300072, China

Corresponding author: Ting Zhang (zhangting hydro@tju.edu.cn)

Dear editor,

Thank you for coordinating the review process and conveying the constructive feedback. We

sincerely appreciate the reviewers' recognition of our manuscript improvements. We have

provided detailed, point-by-point responses to the reviewers' comments in the following pages.

Note that the reviewers' comments are presented in italics, and our responses are in Times New

Roman and blue font. In addition, all the line numbers in the responses refer to the revised

manuscript. All changes made to the manuscript are marked in red font. Please do not hesitate to

contact us if you have any questions or require any additional information. Thank you for your

consideration.

Sincerely,

Ting Zhang

To the **Reviewer #1's** comments, we make the following responses and changes in the manuscript:

1. The authors have addressed all the points raised during the first round of review in a very satisfactory manner, significantly improving the quality and clarity of the manuscript. However, I believe that one important aspect has not yet been fully resolved: the issue concerning the division of the dataset into training and test sets.

Specifically, the selection of events in the two subsets appears to have been carried out according to a systematic but not entirely objective criterion. No analysis has been conducted to evaluate how sensitive the results are to the specific assignment of events to the two groups. The absence of techniques such as cross-validation or bootstrapping makes it difficult to assess the robustness and stability of the model with respect to different dataset partitions.

While I understand the constraints due to limited data availability, I suggest that this limitation be at least acknowledged in the discussion or conclusions. It should be clarified that the presented results refer to a specific configuration of the training and test sets, and that no investigation was carried out on the influence of different event assignments.

Such a clarification would strengthen the scientific approach of the study and provide greater methodological transparency.

Response: Thank you for your insightful comment. We have supplemented the division method between the training set and the validation set in the available dataset. The division of the two datasets took into account the main characteristics of floods, including temporal occurrence, peak discharge, and flood duration, to ensure that both datasets comprehensively cover diverse flood

characteristics as much as possible.

**In the revised manuscript, Page 14, Line 327-332:**

The partitioning of training and validation sets was designed to ensure balanced representation of flood characteristics across both datasets, specifically considering temporal occurrence, peak discharge, and flood duration. This stratification achieves comprehensive inclusion of major, moderate, and minor flood magnitudes while encompassing diverse hydrograph types—including both single-peak and multi-peak events—to maintain hydrological process representativeness.

As for the limitations that may arise from insufficient data, we have supplemented this part in the **Discussion** section and clarified that the results provided are based on the specific flood event classification of the training and validation sets, and we have not studied the impact of different dataset classifications on the results.

**In the revised manuscript, Page 30-31, Line 649-660:**

Furthermore, the dataset was partitioned solely into training and validation sets primarily due to limitations in available historical flood events—only 30 events were utilized, most with relatively short durations. This resulted in a limited sample size and insufficient additional floods for model testing; future data acquisitions will be incorporated to enhance robustness. To maximize coverage of flood diversity and capture spatiotemporal heterogeneity, we partitioned data based on temporal occurrence, peak discharge, and flood duration. This methodology follows established precedents (e.g., Lv et al., 2020; Read et al., 2019; Xie et al., 2021; Jiang et al., 2020) where dual-set partitioning is widely adopted beyond flood forecasting applications. Crucially, our results are contingent upon the specific flood event partitioning of training and validation sets detailed in

Table 1, with no investigation of alternative partitioning impacts. Future research could employ cross-validation or bootstrapping to evaluate model robustness and stability across different dataset divisions.